# Low-Density Reference Fingerprinting SNP Dataset of CIMMYT Maize Lines for Quality Control and Genetic Diversity Analyses

**DOI:** 10.3390/plants11223092

**Published:** 2022-11-14

**Authors:** Jingtao Qu, Alberto A. Chassaigne-Ricciulli, Fengling Fu, Haoqiang Yu, Kate Dreher, Sudha K. Nair, Manje Gowda, Yoseph Beyene, Dan Makumbi, Thanda Dhliwayo, Felix San Vicente, Michael Olsen, Boddupalli M. Prasanna, Wanchen Li, Xuecai Zhang

**Affiliations:** 1Maize Research Institute, Sichuan Agricultural University, Chengdu 611130, China; 2International Maize and Wheat Improvement Center (CIMMYT), El Batan, Texcoco 56237, Mexico; 3Asia Regional Maize Program, International Maize and Wheat Improvement Center (CIMMYT), ICRISAT Campus, Patancheru, Hyderabad 502324, Telangana, India; 4International Maize and Wheat Improvement Center (CIMMYT), Village Market, P.O. Box 1041, Nairobi 00621, Kenya

**Keywords:** tropical maize, CIMMYT maize line (CML), reference SNP dataset, fingerprinting, quality control, heterotic group

## Abstract

CIMMYT maize lines (CMLs), which represent the tropical maize germplasm, are freely available worldwide. All currently released 615 CMLs and fourteen temperate maize inbred lines were genotyped with 180 kompetitive allele-specific PCR single nucleotide polymorphisms to develop a reference fingerprinting SNP dataset that can be used to perform quality control (QC) and genetic diversity analyses. The QC analysis identified 25 CMLs with purity, identity, or mislabeling issues. Further field observation, purification, and re-genotyping of these CMLs are required. The reference fingerprinting SNP dataset was developed for all of the currently released CMLs with 152 high-quality SNPs. The results of principal component analysis and average genetic distances between subgroups showed a clear genetic divergence between temperate and tropical maize, whereas the three tropical subgroups partially overlapped with one another. More than 99% of the pairs of CMLs had genetic distances greater than 0.30, showing their high genetic diversity, and most CMLs are distantly related. The heterotic patterns, estimated with the molecular markers, are consistent with those estimated using pedigree information in two major maize breeding programs at CIMMYT. These research findings are helpful for ensuring the regeneration and distribution of the true CMLs, via QC analysis, and for facilitating the effective utilization of the CMLs, globally.

## 1. Introduction

Maize is one of the most important crops and is widely cultivated all over the world. Maize production of the tropically adapted germplasm occupied 30% of global maize production [1], and it is the preferred staple food for 900 million people living on less than 2 US$ a day, primarily in sub-Saharan Africa, Latin America, and Asia. The International Maize and Wheat Improvement Center (CIMMYT), a non-profit international agricultural research and training organization, seeks to develop improved maize inbred lines and hybrids, with superior performance and multiple stress tolerance, in various agro-ecological zones across Africa, Latin America, and Asia in order to improve maize productivity for resource-constrained smallholder farmers [2]. Between 1984 and 2022, 615 CIMMYT maize lines (CMLs), adapted to the tropical/subtropical maize production environments, had been developed and released as international public goods and were accessible to all. These CMLs represent the broad genetic diversity of improved tropical maize germplasm in the world, and they are freely available to both public and private sector breeders worldwide under the standard material transfer agreement, via the CIMMYT Germplasm Bank (https://www.cimmyt.org/resources/seed-request/#maize (accessed on 13 May 2021)). The development of a reference fingerprinting molecular marker dataset of all currently released CMLs is helpful in order to perform molecular marker-based quality assurance (QA) and quality control (QC) analysis on the increased seed stocks and to assure their genetic purity and identity. In addition, it is also useful for the breeders to better understand how to utilize the currently released CMLs in their breeding programs, based on the genetic diversity analysis and using the reference fingerprinting molecular marker dataset.

The CMLs have been widely requested and used by both public and private institutions all over the world [3]; they are public goods and have significantly contributed to improving abiotic and biotic stress tolerance and grain quality in both temperate and tropical maize [2,4]. The large number of seed requests coming from all over the world each year bring great challenges to the preservation, maintenance, and distribution of the germplasm of CMLs. The QC of seeds in maize breeding and seed production aims to ensure that that the currently used inbred/parental lines are genetically identical to the original source, the seed has required genetic purity, the hybrids are indeed derived from the specified parents, and that the seed lots are meeting all the seed quality standards, etc. [5,6].

Compared to the traditional QC approaches of morphological comparison and isozymes electrophoresis, the molecular marker-based QC—i.e., DNA fingerprinting performed by genotyping at defined molecular markers that cover all 10 chromosomes of maize—is rapid, accurate, cost-effective, and independent to the environment. Over the past several decades, different molecular marker genotyping technologies, including restriction fragment length polymorphisms, simple sequence repeats, and single nucleotide polymorphisms (SNPs), have been applied in DNA fingerprinting and molecular marker-based QC analysis in maize [7,8,9]. Recently, SNPs have become the most promising genotyping technology for DNA fingerprinting, molecular marker-based QC analysis, and database construction, due to their high coverage of the genome, codominance, known chromosomal locations, and the potential for high-throughput and automated analysis. The KBiosciences’ Competitive Allele-Specific PCR system (KASPar; http://www.kbioscience.co.uk (accessed on 1 May 2022)) is a single-plex SNP genotyping platform that has been widely used for molecular marker-based QC analysis, linkage mapping, fine-mapping, and marker-assisted selection in maize [10,11,12]. The KASPar is an ideal SNP genotyping platform for molecular marker-based QC analysis because it can be used to quickly genotype tens of thousands of samples, with tens to hundreds of markers, and it does not require large computing resources and time-consuming bioinformatics analyses, thus allowing breeders to receive the genotypic data and analysis results in a shorter time than other genotyping platforms. In addition, the cost per sample of KASPar genotyping platforms is more cost-effective than chip-based SNP technologies, such as Infinium, GoldenGate, and Affymetrix Axiom [13,14,15,16,17,18,19,20]. A framework of QC genotyping, with a recommended 50 or 100 KASPar SNPs, has been established in several practical maize breeding programs at CIMMYT [5,6]. A QC genotyping approach was also developed to genotype CML1-561 with the DArT-Seq technology [21], and a total of 88,600 SNP markers were generated on each CML, in which 80 and 10 SNPs were selected as the “broad QC” and “rapid QC” sets, respectively. However, the reference fingerprints of all of the released CMLs to date are not available publicly.

The results of numerous germplasm characterization studies have shown that chip-based SNP genotyping platforms are promising and effective for performing genetic diversity analysis in collections of maize germplasms with different genetic backgrounds, although the cost per sample of chip-based SNP genotyping platforms is still high [10]. Three generations of the maize haplotype map (HapMap) had been reported by re-sequencing 27 to 1218 maize inbred lines, representing the global maize genetic diversity [22,23,24]. Genotyping-by-Sequencing (GBS) is one of the most widely used simplified reduced-representation sequencing approaches, it is high-throughput and cost-effective; the genotyping cost per sample was decreased by reducing genome complexity with restriction enzymes, and multiplex sequencing the samples by using the barcode system [25,26,27,28]. A molecular characterization study on all of the CMLs before CML539 was carried out using GBS SNPs to better understand how to utilize these CMLs [2]. However, new CMLs are continuously being released and the genotypic datataset has to be updated to include new lines. A cost-effective genotyping platform with a short turnaround time, such as KASPar, is required to genotype all of the CML candidates before release in order to generate the reference fingerprinting SNP dataset for all CMLs, as a whole. This can then be used to examine the genetic purity and identity of the CML candidates, and will avoid the new CMLs having close genetic relationships with existing CMLs.

In the present study, all of the currently released 615 CMLs (available till 2022) were genotyped with 180 KASPar QC SNPs, routinely used at CIMMYT, and two to five biological replications per CML were genotyped. The main objectives of the present study were to: (1) develop the reference fingerprinting SNP dataset of all currently released 615 CMLs; (2) implement QC analysis on two to five biological replications of each CML in order to identify suspicious CMLs in the CIMMYT gene bank with issues of genetic purity, identity, seed contamination or mislabeling during genotyping; (3) conduct genetic diversity analysis on all of the currently released 615 CMLs and 15 temperate maize inbred lines.

## 2. Results

### 2.1. Summary of KASP Marker Characteristics

All 180 KASP SNP markers were evenly anchored on the reference genome of B73_RefGen_V4, based on their flanking sequence information. In total, 179 SNP markers were anchored on the ten maize chromosomes, and one SNP marker was anchored on the B73_RefGen_V4_Contig104 (Figure 1). The number of SNPs on each chromosome ranged from 12, on chromosome 4, to 22 on chromosome 6, with a mean of 17.9 markers per chromosome (Table 1). The physical distance of two adjacent SNPs on each chromosome varied from 114 bp on chromosome 7, between SNP PZE-101093951 and SNP PZE0186065237, to 58.54 Mb on chromosome 4, between SNP PZE-101093951 and SNP PZE0186065237, with an average value of 12.30 Mb (Appendix A). Among the 180 KASP SNPs, seven SNPs were breeder-ready markers for provitamin A (PVA); three SNPs were breeder-ready markers for resistance to maize streak virus (MSV); 10 SNPs were breeder-ready markers for resistance to maize lethal necrosis (MLN); 10 SNPs were breeder-ready markers for quality protein maize (QPM); and 2 SNPs were breeder-ready markers for resistance to tar spot complex (TSC).

In total, 360 alleles were detected for all the 180 KASP SNPs, with two alleles per SNP, as was expected. The SNP mutation type analysis indicated that 59.9% of the SNP mutations belong to the A/G, T/C type; 20.9% of the SNP mutations belong to the A/C, G/T type; and 19.2% of the SNP mutations belong to the A/T, C/G type. SNP is a type of mutation that occurs in a single nucleotide in the genome. In the present study, the number of KASP markers with the allelic types of A/G and T/C was higher than those with other allelic types. The summary information for all 180 KASP SNPs in the first round of genotyping dataset are shown in Figure 2a. The average heterozygosity rate across all the SNPs was 1.14%, with a range from 0 to 19.73% per SNP. In total, 172 of the 180 SNPs had a heterozygosity rate lower than 5% (Appendix A). The remaining eight SNPs with heterozygosity rates greater than 5%—i.e., C6_9203520_v3_22, PZA02746_2, C3_169075836_v3_22, C2_64580161_v3_22, PZA01919_2, PHM2350_17, C6_151676156_v3_22, and PZA00413_20—were excluded from the further QC analysis and development of the reference fingerprinting SNP dataset. The average missing rate across all the SNPs was 5.55%, with a range from 0.42% to 37.75% per SNP. In total, 164 of the 180 SNPs had their missing rates lower than 10%. The remaining 16 SNPs with missing rates > 10%—i.e., C3_11540103_v3_22, C7_141983951_v3_22, PZA03211_6, C7_158970827_v3_22, PZE-110083653, PZA02746_2, C2_64580161_v3_22, C6_151676156_v3_22, PZA00223_4, C10_132535344_v3_22, C6_167167466_v3_22, C8_114824208_v3_22, C4_239917922_v3_22, PHM13440_13, C5_152469859_v3_22, and C5_86188043_v3_22—were excluded from further QC analysis and development of the reference fingerprinting SNP dataset. The average MAF across all the SNPs was 0.32, with a range from 0 to 0.50. In total, 170 of the 180 SNPs had MAF values greater than 0.05. The remaining 10 SNPs with MAF values lower than 0.05—i.e., PZE-101093951, PZE-110083653, C7_141983951_v3_22, C7_158970827_v3_22, C10_132535344_v3_22, S6_18924381, S10_134655704, S10_134583972, S10_136840485, and S10_137904716—were excluded from further QC analysis and development of the reference fingerprinting SNP dataset. Finally, 152 SNPs, which had heterozygosity rates lower than 5%, missing rates lower than 10%, and MAF values greater than 0.05, were retained and used in the further QC analysis and development of the reference fingerprinting SNP dataset.

The genes associated with the KASP markers were used to perform GO analysis. In total, 164 of the 180 KASP SNP markers were associated with 208 genes, including 151 KASP SNP markers located in the genic region, 15 KASP SNP markers located within the two kb upstream intergenic region of a gene, and 34 KASP SNP markers located within the two kb upstream or downstream intergenic region of a gene. Among the 208 genes, 181 of them were annotated with potential function, such as Zm00001d018971, a regulatory protein of opaque-2 and Zm00001d034385, aldehyde oxidase 3, associated with drought stress tolerance (Appendix A). In total, 204 of the 208 genes were annotated into 134 GO terms, divided into three categories: 66 terms involved in biological processes, 43 GO terms associated with cellular components, and 25 GO terms correlated with molecular functions (Figure 3; Appendix A). In addition, four GO terms were associated with the development and regulation of quality protein maize, including GO:0044267 (cellular protein metabolic process), GO:0006464 (cellular protein modification process), GO:0019538 (protein metabolic process), and GO:0005515 (protein binding).

### 2.2. Quality Control Analysis Results

The genetic purity analysis revealed that the average heterozygosity rate across two biological replications of all the 603 CMLs was 2.29%. In total, 556 of the 1206 samples, i.e., 46.10%, had heterozygosity rates lower than 1%, and the seeds of these samples achieved the required purity level. In addition, 1058 of the 1206 samples, i.e., 87.73%, had heterozygosity rates lower than 5%, indicating that the seeds of these samples achieved the basic purity level. The 151 CMLs with a heterozygosity rate of at least one biological replication greater than 3% were used for the second round of genotyping, and are listed in Appendix A.

The genetic identity analysis revealed that the average similarity rate between the two biological replications across all the 603 CMLs was 98.54%, indicating that most of the CMLs were essentially identical for the two biological replications genotyped in the first round of genotyping. The similarity rates between the two biological duplications of 485 CMLs were greater than 99%, indicating that these CMLs were essentially identical. The similarity rates between the two biological duplications of 93 CMLs were greater than 95%, indicating that these CMLs were identical, with slight genetic divergences caused by genetic drift. The rest of the 25 CMLs had similarity rates lower than 95% between the two independent biological replications, indicating either a genetic identity issue of the seeds or mislabeling during genotyping. Among these 25 CMLs, eleven CMLs had relatively high similarity rates, of greater than 85%, and relatively high heterozygosity rates, of greater than 5% in at least one biological replication; this result indicated that the genetic identity issue of these CMLs, including CML51B, CML95B, CML196A, CML208A, CML437A, CML473B, CML474A, CML487B, CML519B, CML521B, and CML523A, were probably caused by the issue of impurity seeds. The rest of the fourteen CMLs, including CML76A, CML80A, CML82A, CML83A, CML96B, CML100A, CML101A, CML107A, CML111A, CML112A, CML113A, CML116B, CML240A, and CML418B, had similarity rates lower than 85% between the two independent biological replications, and six of these fourteen CMLs, including CML76A, CML96B, CML100A, CML107A, CML111A, and CML240A, had relative high heterozygosity rates of greater than 5%. The genetic identity issue of these six CMLs was most likely caused by the issue of impurity seeds, and the genetic identity issue of the other eight CMLs was most likely caused by mislabeling during genotyping. In these 25 CMLs, the genetic distance between the two biological duplications of a few CMLs was greater than the between different CMLs, and they were selected in the second round of genotyping.

Among the 540 samples of 180 CMLs genotyped in the second round, the heterozygosity rates of 482 samples, corresponding to at least one biological replication of 179 CMLs, were lower than 5%, including 323 samples with heterozygosity rates lower than 1%, and 111 samples with heterozygosity rates between 1% to 3%. The biological replications with heterozygosity rates lower than 3% from both rounds of genotyping were used to build the reference SNP dataset for the fingerprinting and QA/QC analyses. In addition, 13 of the 25 CMLs with similarity rates lower than 95% between the two biological replications in the first round of genotyping were identified with mislabeling issues in one biological replication in the first round of genotyping (Figure 4), based on the neighbor-joining tree results across all five biological replications. The reference fingerprinting SNP dataset for these thirteen CMLs, including CML76A, CML80A, CML82A, CML83A, CML96B, CML100A, CML101A, CML107A, CML111A, CML112A, CML113A, CML116B, CML240A, was built with the reliable genotypic data of the four biological replications from both rounds of genotyping.

### 2.3. Low-Density Reference SNP Dataset for Fingerprinting and QA/QC Analysis

The low-density reference fingerprinting SNP dataset was developed for all of the currently released 615 CMLs through the integration of the reliable genotypic data of all the biological replications for each CML. For CML1A to CML603A, 152 SNPs were used to develop the reference SNP dataset; the SNPs in the first round of genotyping with a heterozygosity rate greater than 5%, a missing rate greater than 10%, and a MAF lower than 0.05, were excluded from the reference SNP dataset. For CML604A to CML615A, only one biological duplication for each CML was genotyped with 85 of the 152 SNPs to develop the reference SNP dataset. In total, the reference SNP dataset included 584 CMLs with a heterozygosity rate lower than 5% and an average similarity rate greater than 95% between the biological replications for each CML; 22 CMLs with a heterozygosity rate lower than 5%, and an average similarity rate lower than 95% between the biological replications for each CML; and CML179A with a heterozygosity rate of 10.08% and an average similarity rate lower of 97.94% between the biological replications (Appendix A). Genotyping data of CML543B, CML569B, CML585A, and CML605B were missing in the current reference SNP dataset. Among the 22 CMLs with heterozygosity rates lower than 5% and average similarity rates lower than 95% between the biological replications, the similarity rates between the biological duplications of eight CMLs were lower than the similarity rate with the other CML.

The summary information of the 152 SNPs in the reference fingerprinting genotypic dataset in all the 584 reliable CMLs is shown in Figure 2b. The number of SNPs per chromosome ranged from 11 on chromosome 4 to 21 on chromosome 1, the average missing rate across all the SNPs was 2.50%, the average heterozygosity rate across all the SNPs was 0.60%, and the average MAF across all the SNPs was 0.34. The 152 SNPs in the reference fingerprinting genotypic dataset had lower missing and heterozygosity rates, and higher MAF than those in the original 180 SNPs dataset. In the reference fingerprinting SNP dataset, the average missing rate across all the reliable 584 CMLs was 2.50%, and the average heterozygosity rate across all the reliable 584 CMLs was 0.60% (Appendix A).

The genetic distance result for the 25 pairs of CML with pedigree-based close relationships is shown in Table 2, where the genetic distance between each pair of CMLs varied from 0.02 to 0.44, with a mean of 0.14. This result indicates that the high-quality reference SNP dataset developed in the present study is appropriate for fingerprinting, QA/QC, and genetic diversity analyses, and it can accurately reflect the relationships of the genetically close related CMLs.

### 2.4. Genetic Diversity Analysis Conducted with the Reference SNP Dataset

The result of the principal component analysis in all of the currently released 615 CMLs, along with the fourteen temperate maize inbred lines, is shown in Figure 5. The first three principal components explained 7.74% of the genetic variations, in total, and the genetic variation explained by the first principal component, the second principal component, and the third component was 3.12%, 2.58%, and 2.24%, respectively. A clear genetic divergence was observed between temperate and tropical inbred lines, while a clear clustering pattern was not observed between the BSSS and NSSS heterotic groups in the temperate maize, due to the small sample size of the temperate subgroup affecting the accuracy of cluster pattern analysis within this subgroup. The Highland Tropical subgroup, on the whole, was separated from the Lowland Tropical and Subtropical/Mid-altitude subgroups, whereas the Lowland Tropical subgroup and the Subtropical/Mid-altitude subgroup overlaid partially with each other.

The pairwise genetic distance between all the 584 reliable CMLs varied between 0 and 0.63, with an average of 0.43 (Appendix A). The results showed that 99.43% of the pairs of CMLs had genetic distances greater than 0.30, including 92.49% that had genetic distances between 0.30 to 0.50, and 6.94% with genetic distances greater than 0.50. Less than 1% of the pairs of CMLs had genetic distances lower than 0.30, including 0.49% ranging between 0.10 and 0.30; 0.08% of them ranging between 0 and 0.10; and ten pairwise of CMLs with genetic distances equal to 0, i.e., CML117A-CML133A, CML138B-CML367B, CML171B-CML172B, CML204B-CML208A, CML214A-CML216A, CML283A-CML284A, CML377B-CML383B, CML387A-CML504B, CML472B-CML474A, and CML563A-CML579A. Of the 86 pairwise of CMLs with genetic distances lower than 0.05, 34 pairwise of CMLs were siblings, and 14 pairwise of CMLs were selected from the same population. This information indicates that most of the lines in the entire panel are either not related or only distantly related to each other and reflects the high genetic diversity of the CML germplasm source. However, it also indicates that the genetic distances between the CML candidates and the existing CMLs should be greater than 0.05 to avoid releasing genetically close-related new CMLs.

The average genetic distances between the different subgroups were calculated. The results indicated that the genetic distance between the tropical and temperate maize was greater than those among the three tropical subgroups. The genetic distance between the Temperate subgroup and each of the three tropical subgroups was 0.51. The genetic distance among each pair of all three tropical subgroups was 0.43 or 0.44.

### 2.5. Heterotic Pattern in Two Major Maize Breeding Programs

The analysis of the heterotic groups for each CIMMYT maize breeding program can facilitate the exchange and utilization of global germplasm resources. The 615 CMLs that are currently available have been divided into heterotic group A (HGA) and heterotic group B (HGB), according to the information of pedigrees and combining abilities. Within the Lowland tropical-Mexico (LLT) and East Africa (EA) CIMMYT maize breeding programs, a collection of key founder lines, including the most frequently used breeding lines and testers, were selected to investigate their genetic relatedness with each other (Figure 6 and Figure 7). For LLT, 18 key founder CMLs, including 11 CMLs belonging to HGA and 7 CMLs belonging to HGB, were used to analyze the heterotic pattern in this breeding program. The average pairwise genetic distance within HGA, within HGB, and between HGA and HGB, were 0.28, 0.40, and 0.43, respectively. Within HGA, the CMLs are primarily derived from the Tuxpeño population, all the CMLs were closely related to each other, with a maximum genetic distance of 0.34, with the exception of two early-developed lines of CML247A and CML254A (Figure 6). Within HGB, fewer CMLs were selected, and most CMLs were distantly related to each other. In the LLT breeding program, the average pairwise genetic distance between the heterotic groups was greater than those within the heterotic groups, and the CMLs from the same heterotic group were more closely related to each other, particularly in HGA; this indicates that the heterotic pattern is being developed in this breeding program through breeding selection and recycling of a core set of key founder line.

For EA, 44 key founder CMLs, including 28 CMLs belonging to HGA and 16 CMLs belonging to HGB, were used to analyze the trends of heterotic patterns in this breeding program. The pedigree-based heterotic pattern was consistent with that estimated using the reference fingerprinting SNP dataset, with the exception of three CMLs mixed between the two heterotic groups, i.e., CML519B, CML544B, CML567A. This indicates that the CMLs from the same heterotic group were more closely related, and heterotic patterns are being developed in EA. In addition, a few key founder lines developed by the LLT breeding program, i.e., CML495A, CML494B, and CML576B, are also being used in EA to broaden the genetic diversity. The heterotic group information of these introduced lines is consistent with that of the EA lines, showing the feasibility of harmonization of heterotic patterns and utilization of founder lines across breeding programs, via the global germplasm exchange.

## 3. Discussion

The collection of currently released CMLs, which represent the diversity of improved tropical maize germplasm, has been distributed to various public and private sectors worldwide to improve a significant number of value-added traits in temperate and tropical maize. The reference fingerprinting SNP dataset for all the currently released CMLs developed in this study is helpful for the CIMMYT germplasm bank; in order to regenerate, maintain, and distribute the CML germplasm worldwide, they can genotype the regeneration seeds with the reference fingerprinting SNP dataset to assess their genetic purity and identity. Seed requesters can also use the reference fingerprinting SNP dataset to evaluate whether the obtained CML seeds are what they expect. Moreover, the reference fingerprinting SNP dataset developed by the present study is also useful for the breeders from both CIMMYT and partner institutes to better understand how to utilize the currently released CMLs in their breeding programs to select parental lines, replace testers, assign heterotic groups and create a core set of breeding germplasm [2].

Previous studies only genotyped part of the currently released CMLs, using different genotyping platforms to perform QC and genetic diversity analyses [2,6,21,29]. To the best of our knowledge, the reference fingerprinting molecular marker dataset had been not developed for all the currently released CMLs with any genotyping platform. In addition, single replication was genotyped for each CML, and a few CMLs were selected as technical replications in the previous studies. In the present study, two rounds of genotyping were applied to generate the reference fingerprinting SNP dataset for all the currently released 615 CMLs, and each CML was genotyped with at least two biological duplications to ensure the reliability of this reference fingerprinting SNP dataset. Using the genotypic data of the five biological duplications from both rounds of genotyping, mislabeled samples corresponding to thirteen CMLs in the first round of genotyping were also identified, and the reference fingerprinting SNP dataset for these thirteen CMLs was improved by removing the genotypic data of the mislabeled duplications. Furthermore, 28 SNPs with heterozygosity rates greater than 5%, missing rates greater than 10%, and MAF values lower than 0.05, were excluded from the reference fingerprinting SNP dataset. For the rest of the 152 SNPs, the average missing was 2.50%, the average heterozygosity rate was 0.60%, and the average MAF was 0.34. These results indicate that these 152 SNPs are high quality, and they are appropriate for further genetic diversity analysis, as well as for future QC studies.

Due to a large number of germplasm requests all over the world, regenerating, maintaining, and distributing the CML germplasms has become a significant challenge. During seed regeneration, there is a possibility of contamination with seeds or pollen [3]. The QC genotyping is important for the regeneration, maintenance, and distribution of CML seeds. The genetic purity analysis of the present study showed that 12.27% of the samples in the first round of genotyping did not achieve the basic purity level. Most of these samples correspond to the early CMLs, developed with the recurrent method of population improvement, and the original seeds of these CMLs submitted to the CIMMYT maize germplasm bank were not purified completely. For preserving the genetic diversity of the CMLs, the bulk pollen method was applied for regeneration seeds, and it also maintained the heterozygosity in the regeneration seeds. The early CMLs, with a possibly high level of heterozygosity rate identified in the present study, need to be purified by the CIMMYT germplasm bank and restock the regeneration seeds. For the CMLs with a possibly high level of heterozygosity rate released in the past two decades, and still being frequently used by the breeding program, the CIMMYT maize breeding team take responsibility for purifying them and re-submitting the purified increased seeds to the CIMMYT germplasm bank. Then, the reference fingerprinting SNP dataset for these CMLs needs to be updated by sampling the purified new seed sources.

Based on the first round of genotypic data, 25 CMLs were identified with the genetic identity issue, and eleven of these CMLs have both identity and purity issues; therefore, these CMLs need to first be purified, and then the reference fingerprinting SNP dataset of these CMLs need to be updated by sampling the purified new seed sources. Among these 25 CMLs, the genetic distance of seven pairs of CMLs, estimated with the reference fingerprinting SNP dataset, was zero, and most of these pairs of CMLs had similar genetic backgrounds and closer pedigree relationships. These results indicated that the CMLs with close pedigree relationships could not be distinguished by the low-density reference fingerprinting dataset of 152 SNPs and that, before release, all the CMLs candidates have to be genotyped with the same reference fingerprinting dataset as that of the currently released CMLs. In addition, the genetic relationships between the CML candidates and all currently released CMLs need to be estimated to avoid releasing any new CMLs that have close genetic relationships with the existing CMLs, and the seed purity of the CML candidates should be tested before being submitted to the CIMMYT maize germplasm bank for release. Moreover, the reference fingerprinting SNP dataset for all the CMLs can be updated as soon as the new CMLs are announced for release.

The results of the genetic diversity analysis performed in the present study showed that the average genetic distance between all paired CMLs was 0.43, and 99.43% of pairs of CMLs had genetic distances greater than 0.30, indicating that high genetic diversity existed in the collection of the currently released CMLs, and most CMLs are distantly related. These results are similar to the previous studies [2]. The result of the principal component analysis and the average genetic distances between the different subgroups revealed that a clear genetic divergence was observed between temperate and tropical maize; the Highland Tropical subgroup can be distinguished from the Lowland Tropical and Subtropical/Mid-altitude subgroups. The Lowland Tropical subgroup and the Subtropical/Mid-altitude subgroup partially overlapped with each other; this reflects the current germplasm exchange flow from the Lowland Tropical breeding programs to the Subtropical/Mid-altitude breeding programs. The currently released CMLs are clustered according to their environmental adaptation groups and pedigree relationships.

The heterotic patterns in maize are discovered based on whether phylogenetic and geographical isolation exists, as well as being enhanced by breeders through long-term selection [29]. The heterotic groups within a collection of maize germplasm can be estimated based on pedigree information and by combining ability tests, genetic distance, or kinship coefficient analysis. Using the molecular markers provides an alternative approach for revealing the heterotic patterns in a collection of maize germplasm. A previous study showed that only half of CMLs have heterotic information [2], which was provided by the individual breeding program based on the information of pedigree and combining ability. In the present study, the heterotic group information for all of the currently released CMLs was assigned to specific heterotic groups of CIMMYT: A and B. This will facilitate the germplasm exchange and utilization among the different maize breeding programs of CIMMYT. The pairwise genetic distances between all of the currently released CMLs were also estimated based on the reference fingerprinting SNP dataset, which is complementary to the updated heterotic group information to help breeders to better understand the utilization of the currently released CMLs in their breeding programs. In two major maize breeding programs at CIMMYT, the pedigree-based heterotic pattern of the important CMLs, widely used in breeding and tests, were mostly consistent with those estimated using the reference fingerprinting SNP dataset. There were only a few exceptions of inbred lines, mixed between the two heterotic groups, indicating that the CMLs from the same heterotic group were more closely related, and heterotic patterns are being developed through breeding selection and recycling of a core set of key founder lines. The correct parental combinations to maximize heterosis can be identified to develop high yield potential hybrids.

## 4. Materials and Methods

### 4.1. Plant Materials

In the present study, 615 CMLs and 14 temperate maize lines were used for genotyping in order to perform QC and genetic diversity analysis. Seeds of 615 CMLs were obtained from the CIMMYT Maize Germplasm Bank. The modified CTAB (cetyltrimethylammonium bromide) method was applied for DNA extraction from the leaf tissues [30]. All of the CMLs were planted at the Tlaltizapan experimental station of CIMMYT, with 25 seeds per line per plot. The leaf tissues of each CML were sampled by mixing 10 individual plants, and two independent biological replications were collected for each inbred line. For the second round of genotyping, a subset of 180 CMLs was selected; the heterozygosity rate of at least one biological duplication of these CMLs was greater than 3%, and the genetic distance between the biological replications of the same CML was greater than that between different CMLs. In the second round of genotyping, these 180 CMLs were planted in the greenhouse for tissue collection, a leaf sample from a single plant was collected for each independent biological duplication, and three independent biological duplications were collected for each CML. The latest batch of CMLs (CML604 to CML615), released in May 2021, was genotyped with a subset of 90 SNPs, and each CML was sampled by mixing three individual plants.

Based on the environmental adaptation, all of the 615 CMLs developed by the different breeding programs were classified into three categories: 336 CMLs adapted to the lowland tropical environment, 230 CMLs adapted to the subtropical/mid-altitude environment, and 37 CMLs adapted to the highland tropical environment (Table 3). The maize breeding programs in Latin America released 464 CMLs, including 285 lowland topical CMLs, 144 subtropical/mid-altitude CMLs, and 35 highland tropical CMLs. The maize breeding programs in Southern and Eastern Africa released 110 CMLs, including 22 lowland topical CMLs, 86 subtropical/mid-altitude CMLs, and 2 highland tropical CMLs. The maize breeding program in Asia released 29 lowlands topical CMLs (Table 3). To increase the utilization of the CMLs in maize breeding programs globally, all the CMLs in the present study had been assigned to specific heterotic groups of CIMMYT: A and B, according to the information of pedigrees and combining abilities (personal communication, Felix San Vicente, 2021). The heterotic group assignment is represented for each CML after the CML number, for example, CML312A or CML444B.

In total, 14 temperate maize inbred lines requested from the US Department of Agriculture (USDA)-Agricultural Research Service (ARS) North Central Regional Plant Introduction Station (NCRPIS) in Ames, Iowa, were used in the present study as controls to assess the genetic divergence between temperate and tropical maize. These inbred lines were primarily the Germplasm Enhancement Maize (GEM) accessions and the maize inbred lines, with expired plant variety protection. The fourteen temperate maize inbred lines, including 11 inbred lines from the Stiff Stalk Synthetic (BSSS) heterotic group, and 3 inbred lines from the non-Stiff Stalk Synthetic (NSSS) heterotic group, were referred to as the Temperate subgroup in the present study.

### 4.2. KASP SNP Markers Used for Genotyping

A set of over 1200 KASPar SNP genotyping assays was developed from the mapping data against the B73 reference genome [31] and a SNP mining study from EST sequences [32], which is now publicly available to the global maize genetics community, and it has been routinely applied at the Global Maize Program of CIMMYT for QA/QC, fingerprinting analysis, marker-assisted selection (MAS), and genetic diversity analysis. In total, 180 SNPs prioritized by CIMMYT for QC analysis were selected for genotyping all the currently released 615 CMLs and the fourteen temperate maize inbred lines in the present study (Appendix A). The 180 SNPs were selected because of their high polymorphic information content (PIC), high minor allele frequency (MAF), and uniform coverage of the maize genome. Among these 180 SNPs, subsets of 50 or 100 SNPs were used in previous studies as the core set for routine QA/QC analysis [6], and dozens of KASPar SNPs converted from the DArT-Seq platform were developed for “broad” or “rapid” QA/QC analysis of CMLs [21]. In addition, some breeder-ready markers for implementing MAS were also converted into the KASPar SNPs and included in this marker set. These breeder-ready markers are associated with several traits for tropical maize improvement (https://excellenceinbreeding.org/toolbox/tools/kasp-low-density-genotyping-platform (accessed on 1 May 2022)) [33], including resistance to maize streak virus [12], resistance to maize lethal necrosis [34], kernel provitamin A content [35], quality protein maize [36], and resistance to tar spot complex [11,37] etc.

### 4.3. Characteristics of the SNPs Used for Genotyping

The summary statistics of each SNP in the genotypic dataset across 1206 samples of CMLs in the first round of genotyping, including PIC, MAF, missing rate, and heterozygosity rate, was analyzed using an in-house Perl script. To evaluate the uniformity of distribution, 180 SNPs were anchored to maize reference genome B73_RefGen_v4 using BLAST software and an in-house Perl script [38,39]. The genes harboring the SNP or in a flanking region of two Kb were extracted and annotated using gene functional information, downloaded from the MaizeGBD database [40]. The physical position of the SNPs on the reference genome, and the candidate genes nearby these QC SNPs and their functions, are shown in Appendix A.

The gene ontology (GO) functional analysis was performed with the genes harboring the KASP markers using the singular enrichment analysis (SEA) with the online software AgriGO (http://bioinfo.cau.edu.cn/agriGO/index.php (accessed on 1 May 2022)) [41]. In the GO enrichment analysis, the reference gene set of “Maize AGPv4 (Maize-GAMER)” was chosen as the background and the “plant GO slim” was used as the gene ontology type. The *p*-value was adjusted using the hypergeometric test with a Benjamini-Hochberg false discovery rate (FDR). The visualization of GO enrichment was performed using the ggplot2 package in R.

### 4.4. Quality Control Analysis

The QC analysis was performed on 603 CMLs. First, the missing rate and the heterozygosity rate, representing the genetic purity, were calculated on each biological replication for CMLs using the in-house Perl scripts (www.perl.org (accessed on 1 May 2022)). The genetic purity of each biological replication for all the CMLs was classified into three levels by their heterozygosity rates: purity level with a heterozygosity rate of less than 1%; basic purity level with a heterozygosity rate ranging from 1% to 5%; and heterozygosity with a heterozygosity rate greater than 5%. The CMLs with a heterozygosity rate of at least one biological replication greater than 3% were included in the second round of genotyping, with the leaf tissue collected from the individual plant and three biological replications per CML. For these CMLs, the reference fingerprinting SNP dataset was obtained by integrating the fingerprinting genotypic data of each CML from the five independent biological replications.

Second, the genetic identity analyses were conducted between the two independent biological replications and among all the CMLs by evaluating the similarities between the two replications for the same CML or between the different CMLs. The similarity rate between the two replications or the two genotypes was calculated as the total number of identical alleles divided by the total number of non-missing alleles. The maximum identity value is 1, and the minimum identity value is 0. For the CML with the genetic distance between the two biological replications greater than that between different CMLs, the genotypic data from all the five independent biological replications was used to remove the mislabeled genotyping samples, and develop the reference fingerprinting SNP dataset.

Third, the parent-offspring tests were performed on 25 pairs of CMLs with close relationships to test the accuracy of the reference fingerprinting SNP dataset. Some CMLs are converted from the existing CMLs through marker-assisted selection or backcrossing to improve some specific target traits, such as QPM, imidazolinone resistance for controlling Striga, MSV resistance, and drought tolerance. The genetic distances between the original CMLs and the corresponding converted CMLs were calculated with TASSEL V5.0 software [42].

### 4.5. High-Quality and Low-Density Reference SNP Dataset for Fingerprinting

The reference SNP dataset for all the currently released 615 CMLs was obtained by integrating the fingerprinting genotypic data of each CML from the two independent biological replications in the first round of genotyping. The criteria for integration were as follow: (1) the SNP site containing the consistent alleles in both biological replications, was merged directly; (2) the SNP site containing homozygous alleles in one biological replication and heterozygous alleles in the other biological replication kept the homozygous alleles; (3) the rest of SNP sites were set as missing. The missing rate and heterozygosity rate of each CML across the reference dataset were calculated using the in-house Perl scripts. The SNPs with heterozygosity rates greater than 5%, the missing rate greater than 10%, and the MAF (minor allele frequency) values lower than 0.05 were excluded to develop the reference fingerprinting SNP dataset and to perform further genetic diversity analyses.

### 4.6. Genetic Diversity Analysis

Principal component analysis (PCA) was performed with all of the reference fingerprinting SNPs, on all tropical and temperate maize inbred lines genotyped in the present study, using the TASSEL V5.0 software. The first three principal components were plotted using the Scatter3dplot package in R to show the genetic relationships among the subgroups adapted to different environments.

To reveal the genetic relatedness among all CMLs and the temperate maize inbred lines, the identity by state (IBS) genetic distances were estimated with all the reference fingerprinting SNPs in the TASSEL V5.0 software. The average pairwise genetic distances were also estimated within each subgroup, adapted to different environments, i.e., Lowland Tropical subgroup, Subtropical/Mid-altitude subgroup, Highland Tropical subgroup, and Temperate subgroup.

Based on the pedigree information and the breeding knowledge, a core set of CMLs used by the Lowland Tropical maize breeding program in Mexico and the Subtropical maize breeding programs in East Africa, including the most frequently used CMLs and testers, was selected to investigate the heterotic patterns developed by Lowland tropical-Mexico (LLT) and East Africa (EA) CIMMYT maize breeding program. The phylogenetic tree was constructed based on IBS’s genetic distance matrix for core CMLs in EA using the neighbor-joining algorithm in the ape package in R. The visualization of the phylogenetic tree was performed by MEGA7 software [43].

## 5. Conclusions

In the present study, a set of 180 KASP SNPs was used to genotype all the currently released 615 CMLs to perform QC analysis, build the reference fingerprinting SNP dataset, and conduct the genetic diversity analysis. The QC analysis identified 25 CMLs having purity, identity, or mislabeling issues, further field observation, purification, and re-genotyping work on these CMLs are required. The reference fingerprinting SNP dataset was developed for all the currently released CMLs with 152 high-quality SNPs from 180 KASP SNP markers. Results of principal component analysis and average genetic distances between subgroups showed a clear genetic divergence between temperate and tropical maize, whereas the three tropical subgroups overlaid partially with each other. More than 99% of pairs of CMLs had genetic distances greater than 0.30, showing their high genetic diversity, and most CMLs distantly related. Heterotic patterns estimated with the molecular markers are consistent with those estimated with pedigree information in two major maize breeding programs at CIMMYT. These research findings are helpful to ensure the regeneration and distribution of the true CMLs via QC analysis, and facilitate the effective utilization of the CMLs globally.

## Figures and Tables

**Figure 1 plants-11-03092-f001:**
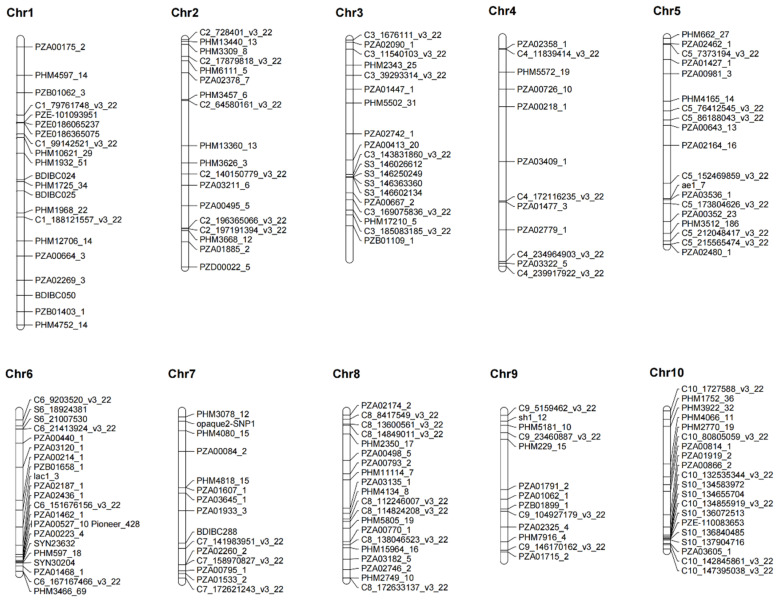
The distribution of 180 SNPs on the ten maize chromosomes.

**Figure 2 plants-11-03092-f002:**
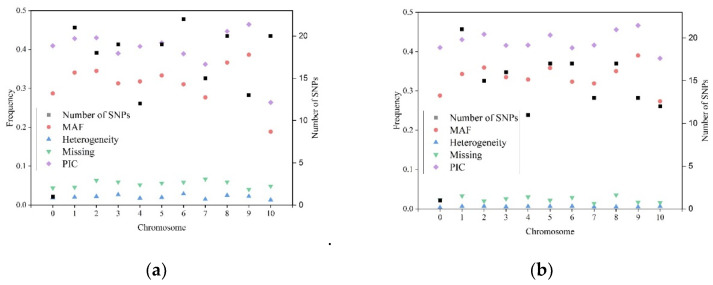
Summary of the heterogeneity, minor allele frequency (MAF), missing rate, and polymorphic information content (PIC) of (**a**) 180 SNPs across 1200 samples and (**b**) 152 high-quality SNPs across 603 CMLs. Chromosome assignments are indicated; where position anchored on non-chromosome, the chromosome is designated as “0”; The heterogeneity, MAF, percentage of Missing value, PIC was shown in left *y*-axis, the number of SNPs for each chromosome was shown in right *y*-axis.

**Figure 3 plants-11-03092-f003:**
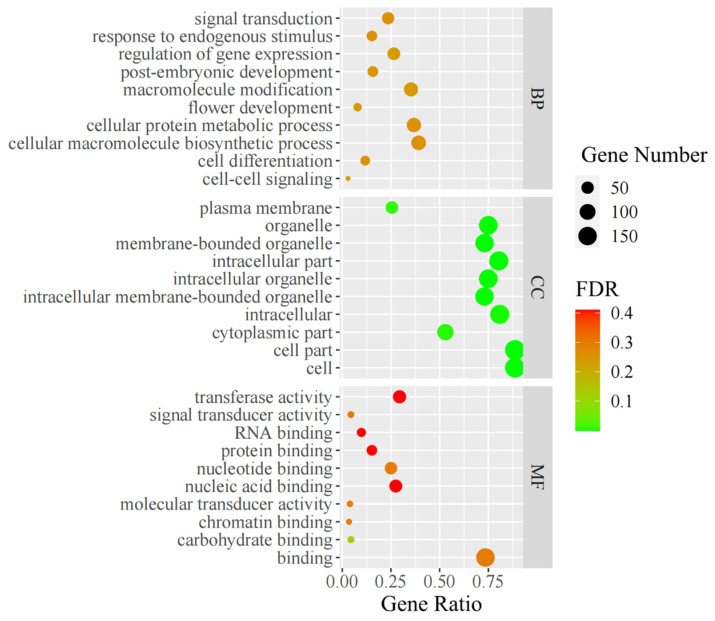
GO enrichment of genes associated with KASP markers. The top 10 of each GO class were shown. BP indicated the class of biological processes, CC indicated the class of cellular components, and MF indicated the class of molecular functions.

**Figure 4 plants-11-03092-f004:**
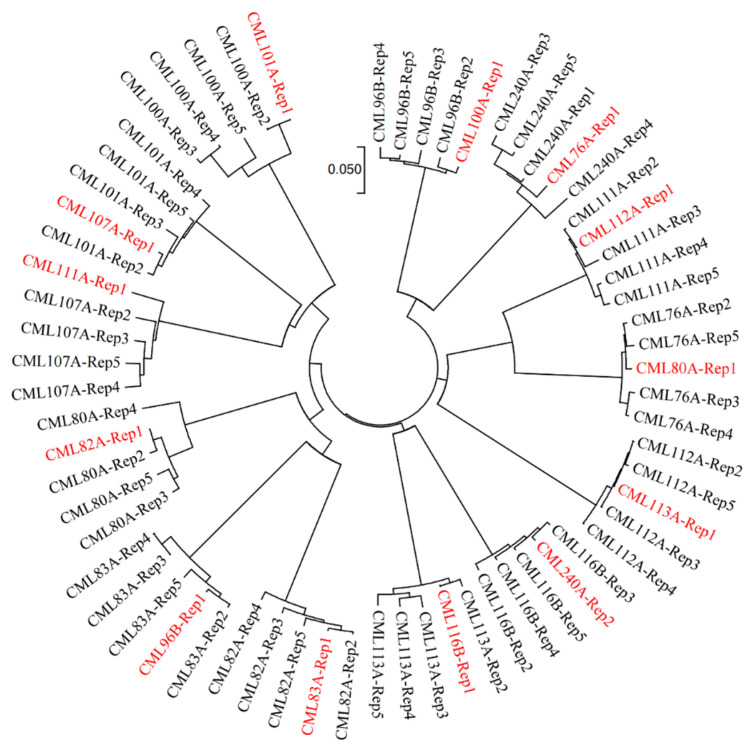
Neighbor-joining tree for the thirteen mislabeled CMLs in the first round of genotyping, estimated with the fingerprinting SNP dataset of five biological duplications from both rounds of genotyping. Rep1 and Rep2 represent the two biological duplications from the first round of genotyping, and Rep3, Rep4, and Rep5 represent the three biological duplications from the second round of genotyping. Mislabeled biological duplications were indicated in *red* color.

**Figure 5 plants-11-03092-f005:**
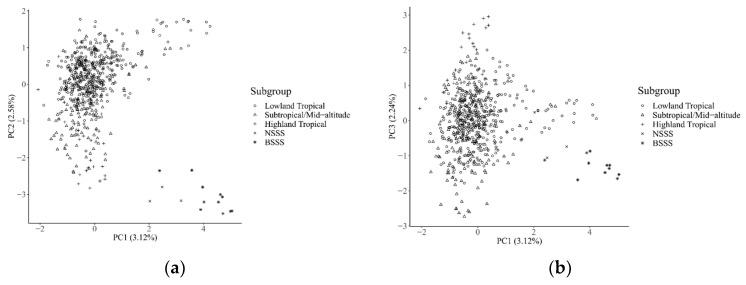
The scatter plot of the first three principal components for 584 reliable CIMMYT inbred lines according to adaptation: Lowland Tropical, Subtropical/Mid-Latitude, and Highland Tropical, and Temperate subgroup of 14 inbred lines belonging to BSSS and NSSS heterotic groups. (**a**) indicated PC1 versus PC2, and (**b**) indicated PC1 versus PC3.

**Figure 6 plants-11-03092-f006:**
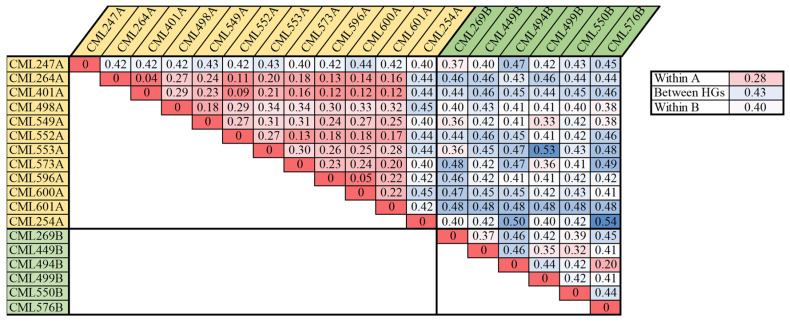
Heat map of the pairwise genetic distances between the 18 key founder CMLs of the Lowland tropical-Mexico (LLT) breeding program, 11 CMLs belonging to Heterotic Group “A”, and 7 CMLs belonging to Heterotic Group “B”.

**Figure 7 plants-11-03092-f007:**
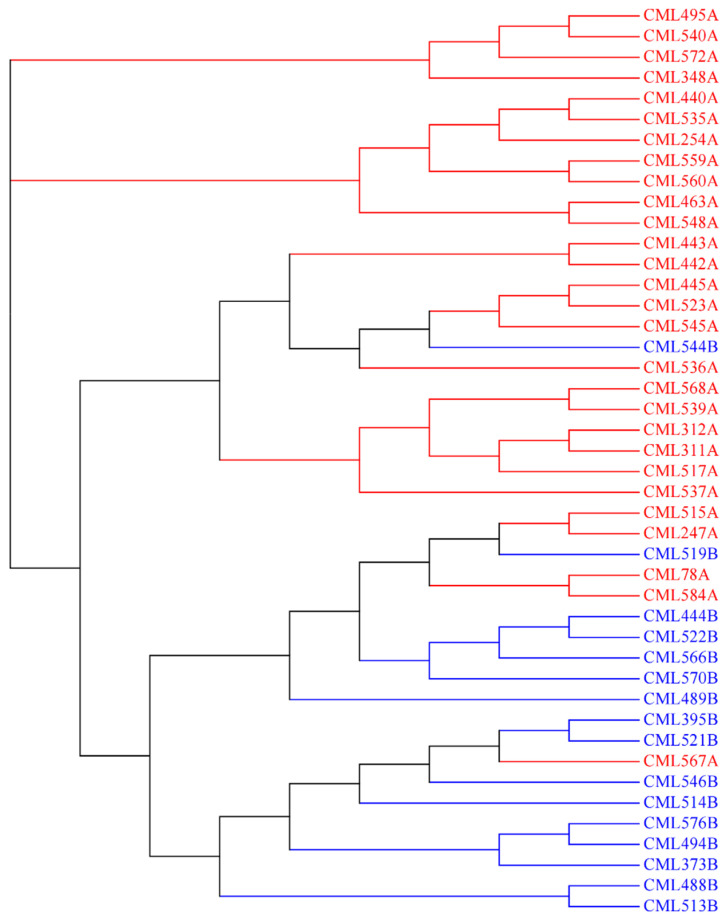
Neighbor-Joining tree based on IBS’ genetic distance for the 44 key founder CMLs of the East Africa (EA) maize breeding program. CMLs from heterotic groups A and B are represented by *red* and *blue*, respectively.

**Table 1 plants-11-03092-t001:** Summary statistics for the 180 KASP markers in the first round of genotyping.

Chromosome	No. of Markers	Minor Allele Frequency	Heterozygosity Rate (%)	Missing Rate (%)
1	21	0.34	1.16	2.27
2	18	0.35	1.18	3.52
3	19	0.32	1.49	3.77
4	12	0.32	1.07	2.44
5	19	0.34	0.86	2.76
6	22	0.31	2.07	3.07
7	15	0.28	0.68	4.77
8	20	0.37	1.09	3.62
9	13	0.39	0.79	2.41
10	20	0.19	0.73	2.52
Ctg104	1	0.29	0.51	2.83
Total	180	-	-	-
Avg.	17.90	0.32	1.14	3.11

**Table 2 plants-11-03092-t002:** The genetic distance between the original parent and the converted line.

Original Parent	Converted Line	Relationship	Genetic Distance
CML78A	CML512A	CML78A IR ^a^	0.05
CML202B	CML513B	CML202B IR	0.44
CML204B	CML514B	CML204A IR	0.04
CML247A	CML515A	CML247A IR	0.08
CML254A	CML516A	CML254A IR	0.08
CML312A	CML517A	CML312A IR	0.18
CML373B	CML518B	CML373B IR	0.05
CML384B	CML519B	CML384B IR	0.05
CML390A	CML520A	CML390A IR	0.15
CML395B	CML521B	CML395B IR	0.03
CML444B	CML522B	CML444B IR	0.09
CML445A	CML523A	CML445A IR	0.02
CML264A	CML503A	CML264A Q ^b^	0.04
CML242A	CML524A	CML242A Q	0.04
CML244A	CML525A	CML244A Q	0.07
CML246B	CML526B	CML246B Q	0.10
CML349A	CML527B	CML349A Q	0.41
CML352A	CML528A	CML352A Q	0.07
CML354A	CML529A	CML354A Q	0.08
CML312A	CML537A	MAS(CML206A/CML312A)-23-2-1-1-B	0.31
CML312A	CML539A	MAS(MSR/CML312A)-117-2-2-1-B	0.20
CML444B	CML566B	(LAPOSTASEQ-C7-F96-1-2-1-1-B*3/CML444B//CML444B)-DH16-B	0.11
CML539A	CML567A	(LAPOSTASEQ-C7-F71-1-2-1-2-B*3/CML539A//CML539A)-DH3-B	0.42
CML539A	CML568A	(LAPOSTASEQ-C7-F71-1-2-1-2-B*3/CML539A//CML539A)-DH20-B	0.19
CML444B	CML570B	(LAPOSTASEQ-C7-F71-1-2-1-2-B*3/CML444B//CML444B)-DH49-B	0.14

^a^ IR: imidazolinone resistant. ^b^ Q: quality protein maize. * indicated harvest with bulk method.

**Table 3 plants-11-03092-t003:** Environmental adaptation, geographic subset, and phenotypic characterization information of 615 CMLs and 14 temperate maize inbred lines.

Environmental Adaptation/ Geographic Subset	No. of Lines	Grain Color		Grain Texture				QPM
		Y	W	D	SD	F	SF	
Mexico Subtropical	146	34	112	51	38	36	21	22
Africa Lowland	22	8	14	6	0	16	0	0
Latin Am Lowland	14	4	10	1	3	6	4	4
Highland Tropical	36	6	30	2	18	3	13	6
Asia Lowland	33	31	2	3	9	17	4	0
Mexico Lowland	249	96	153	69	51	98	31	38
Africa Mid-altitude	93	0	92	9	32	23	35	0
South America	22	17	5	1	2	13	6	0
Temperate	14							

## Data Availability

The dataset presented in this study can be found online at https://hdl.handle.net/11529/10548818 (accessed on 2 November 2022).

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
