# Peer review of "Low-Density Reference Fingerprinting SNP Dataset of CIMMYT Maize Lines for Quality Control and Genetic Diversity Analyses"

_plants, 2022, doi:10.3390/plants11223092_

Round 1

Reviewer 1 Report

Maize is one of the most popular crops worldwide. The article is of interest for solving the problems of maize breeding, creating new hybrids adapted for cultivation on different continents. The authors also develop modern and relatively inexpensive molecular genetic methods for assessing the quality of plant hybrids, improving resistance and increasing crop yields. This is an extremely relevant topic, given the growing population of the planet, the changing climate and the lack of food in many countries of the world. The authors are presented with a number of questions and comments to improve the work.

1. The quality of figure 1 is extremely low and does not allow you to understand what is shown on it. It is recommended to increase the resolution and size of the figure 1.

2. In lines 142-144, the authors indicate the percentage of identified mutations. Explanations or suggestions need to be made for both the causes of mutations and the difference between the types of mutations.

3. Line 145. Indicated in Fig. 2a, but there is no comparison with Fig. 2b.

4. Rows 314-318 show differences in genetic distance between variants, but do not explain its possible causes.

5. In general, throughout the article, it is necessary to look and where it is possible to give explanations for the identified patterns, and not just present the results.

6. Usually in scientific articles the following classical structure of sections is used: Introduction, materials and methods, results, discussion, conclusions. In this article, the materials and methods section is presented at the end. This looks rather illogical.

Reviewer 2 Report

This manuscript presents an interesting study on one SNP dataset of CIMMYT maize lines and its use for control quality and diversity analyses.

In general, the manuscript is well written and could be the interest for the readers.

Some of the figures should be resized, because in the actual size are very difficult to visualize. These figures are: 1, 2, 4 and 5.

The table 3 could be deleted; the data showed here are satisfactory commented in the text (L314-318).

Round 2

Reviewer 1 Report

The authors corrected all the comments made.